# Development of diagnostic and point of care assays for a gammaherpesvirus infecting koalas

**Belinda R. Wright**[1]☉*, **Martina Jelocnik**[2]☉, **Andrea Casteriano**[1], **Yasmine S. S. Muir**[1], **Alistair R. Legione**[3], **Paola K. Vaz**[3], **Joanne M. Devlin**[3], **Damien P. Higgins**[1]

**1** Koala Health Hub, Sydney School of Veterinary Science, University of Sydney, Camperdown, New South Wales, Australia, **2** Centre for Bioinnovation, University of The Sunshine Coast, Sippy Downs, Queensland, Australia, **3** Asia Pacific Centre for Animal Health, Melbourne Veterinary School, Faculty of Science, University of Melbourne, Parkville, Victoria, Australia

☉ These authors contributed equally to this work.
* belinda.wright@sydney.edu.au

**Data Availability Statement:** All relevant data are within the paper and its Supporting Information files.

**Funding:** This study was funded by the Australian Department of Agriculture, Water and the

## Abstract

The recent listing of koala populations as endangered across much of their range has highlighted the need for better management interventions. Disease is a key threat to koala populations but currently there is no information across the threatened populations on the distribution or impact of a gammaherpesvirus, phascolarctid gammaherpesvirus 1 (PhaHV-1). PhaHV-1 is known to infect koalas in southern populations which are, at present, not threatened. Current testing for PhaHV-1 involves lengthy laboratory techniques that do not permit quantification of viral load. In order to better understand distribution, prevalence and impacts of PhaHV-1 infections across koala populations, diagnostic and rapid point of care tests are required. We have developed two novel assays, a qPCR assay and an isothermal assay, that will enable researchers, clinicians and wildlife managers to reliably and rapidly test for PhaHV-1 in koalas. The ability to rapidly diagnose and quantify viral load will aid quarantine practices, inform translocation management and guide research into the clinical significance and impacts of PhaHV-1 infection in koalas.

## Introduction

Koala populations across the Australian states of New South Wales (NSW), Queensland (Qld) and the Australian Capital Territory (ACT) have recently been listed as endangered [1], with disease being a key threat [2, 3]. *Chlamydia pecorum* infections are highly prevalent across koala populations [4], with ocular and urogenital infections potentially leading to blindness, urinary tract disease, infertility or host death [5]. Koala retrovirus (KoRV) is another pathogen infecting koalas that has been associated with chlamydial infections and neoplasia [6, 7]. Two recently discovered gammaherpesviruses, phascolarctid gammaherpesvirus 1 (PhaHV-1) [8] and phascolarctid gammaherpesvirus 2 (PhaHV-2) [9], infect koalas, with research on distribution and prevalence currently limited to South Australian and Victorian koala populations

Environment Bushfire Recovery Multiregional Species Program. BRW is supported by the NSW Wildlife Information Rescue and Education Service (WIRES). The funders had no role in study design, data collection and analysis, decision to publish, or preparation of the manuscript.

**Competing interests:** The authors have declared that no competing interests exist.

[10, 11]. Gammaherpesviruses are one of three subfamilies of double-stranded DNA viruses, *Herpesviridae*, along with the *Alpha-* and *Betaherpesvirinae*. Members of the *Herpesviridae* are well known for establishing latency, with the site of their latent infections being one of the distinguishing features of the three subfamilies [12]. In many host species gammaherpesviruses establish latent infections in B and T lymphocytes with active infections occurring in epithelial cells [13, 14]. Stress and immunosuppression may activate latent gammaherpesvirus infections and this can result in lymphoproliferative disorders in immunocompromised individuals [15, 16]. While the latent site of PhaHV infection has not been confirmed, it has been detected in koala liver, spleen, bone marrow and lymph nodes [17], suggesting lymphocytes as a likely candidate.

While the pathogenesis of the PhaHVs is not yet understood, herpesviruses have been identified as a potential risk to koala health and welfare [3]. This is in part due to the impact of co-infections which may be significant as some co-infections can impact disease progression. For example, research using mouse models has demonstrated that acute infection with a gamma-herpesvirus can inhibit anti-malarial immune responses potentially leading to death [18]. In human studies latent infection by both alpha- and beta-herpesviruses can be stimulated to replicate or to infect normally non-susceptible cells by *Chlamydia trachomatis* infection, and was also able to induce persistence of *C. trachomatis in vitro* [19–21]. One human case study has also found a potential association between co-infection with a gammaherpesvirus and *C. pneumoniae*, and the development of cutaneous lymphoma [22]. In koalas, significant associations have been identified between PhaHVs and detection of *C. pecorum* [10, 23], as well as between PhaHV-1 infection and presence of koala retrovirus (KoRV) [10]. These interactions suggest that immune status and co-infections may be playing a role in disease progression in koalas. For precautionary risk management in koala rehabilitation facilities, PhaHV-1 is considered of higher priority due to its association with KoRV, and its association with older koalas suggesting it is horizontally transmitted, while PhaHV-2 may be acquired early in life through mother to offspring transmission [10, 11].

Understanding and mitigating any role of gammaherpesvirus infection in koala health requires new sensitive and specific diagnostic tests that can detect active infection and quantify viral load, as well as provide rapid results to inform segregation of animals in rehabilitation facilities. In South Australian populations, PhaHVs were detected in 72% of koalas, but detection of the virus from mucosal sites [mostly likely representing active rather than latent infection] was only detected in 54% of koalas [11]. PhaHV infection is typically determined using urogenital swabs [10, 11], possibly because these are routinely collected for chlamydial testing, though the virus is also detectable in oropharyngeal and ocular swabs [10]. Currently, diagnostic testing for PhaHVs involves a nested PCR of the *dpol* gene encoding the viral DNA polymerase catalytic subunit followed by high-resolution melt (HRM) curve analysis [10], which is a laborious process and not suited to quantification, rapid diagnostics or in-clinic settings. Quantitative PCR (qPCR) is considered the gold standard diagnostic assay due to sensitivity and specificity and provides real-time pathogen detection as well as an estimate of viral load [24]. Loop-mediated isothermal amplification (LAMP) is a low-cost rapid pathogen detection method [25] that can be run on a portable instrument, making LAMP highly suitable for both field and in-clinic point of care testing, where immediate decisions are required to limit potential transmission. The utility of chlamydial rapid isothermal testing, including rapid sample processing, has been demonstrated in koala clinical practice to detect *C. pecorum* infections [26, 27].

This study aimed to develop and validate a qPCR assay for use in research settings to detect and quantify load of PhaHV-1 infection. Additionally, we developed and evaluated a LAMP assay as a rapid diagnostic tool for use in point of care testing in clinical and field settings.

These assays have complimentary utility across different settings, allowing research laboratories and veterinary clinics to work collaboratively in gaining vital information on distribution, prevalence and impacts of PhaHV infection.

## Methods

### Samples used for qPCR development and evaluation

Three known positive PhaHV-1 DNA samples extracted from koala swabs from a previous study [10], were used in qPCR assay development and validation. Additional DNA samples were obtained from freshly extracted koala swab samples kept at the Koala Health Hub (University of Sydney). Swab samples were taken from multiple anatomical sites to determine the best sample type for PhaHV-1 detection using qPCR. Sampling of koalas was conducted under the University of Sydney Animal Ethics Approval Number 2021/1975, NSW NPWS Scientific License SL102379 and Qld NPWS WA0019256. We tested a total of 100 swab samples taken on the same date from 47 koalas. Paired oropharyngeal (n = 47) and urogenital (n = 47) swabs were taken from the same koalas as well as ocular (n = 6) swabs from a subset of the same koalas. DNA from swabs was extracted using the Kingfisher MagMAX™ CORE Nucleic Acid Extraction Kit (ThermoFisher Scientific, Australia). Additionally, scat samples were obtained from a subset of 29 of the same koalas on the same date as swab collection.

### DNA extraction from scats

Single koala scat pellets were collected in individual 5 mL tubes containing 3 mL of filter-sterilised lysis buffer (100 mM Tris-HCl; 50 mM EDTA; 1% w/v SDS; pH8) and frozen at -20˚C until DNA was extracted. DNA extraction was carried out using ISOLATE II Fecal DNA Kit (Meridian Bioscience®, Australia) according to the manufacturer's instructions except for the following modifications: Lysis buffer (700 μL) and approximately 200 mg of scat from the collection tube were added to a Bashing Bead Lysis Tube and homogenized at 6 m/s for 30 s using a FastPrep®-24 instrument. Prior to samples being loaded onto a Spin II A Filter, an added centrifugation step (10 000 x g, 3 min) was carried out to pellet colloidal material using a 1.5 mL micro-centrifuge tube. DNA was eluted to a final volume of 100 μL in RNase-free water.

### Primer design

The *dpol* genes of both PhaHV-1 and PhaHV-2 (GenBank accession numbers: JN585829.1, JQ996387.1 respectively) were investigated for appropriate qPCR priming sites. Primers were designed using primer3plus [28], checked for dimerization with Oligoanalyzer [29], and specificity using NCBI BLAST [30]. Multiple primer sets were tested for PhaHV-1 and the primer set with the best amplification was chosen with a resulting product size of 103 bp (Table 1). The sequence available for PhaHV-2 had no appropriate priming sites due to GC content and repetitive bases in the target region as confirmed by testing multiple primer sets for PhaHV-2 *in silico* and in laboratory tests.

**Table 1. qPCR primers for PhaHV-1 developed in this study.**

| Primer | Sequence 5'-3' | Length | Tm | GC % |
|---|---|---|---|---|
| PhaHV-1_fwd | GGGAAGAACTATGTTGGAACGC | 22 bp | 59.6 C | 50.0 |
| PhaHV-1_rev | TGAGTCCTTTTCGCTTGGGA | 20 bp | 59.2 C | 50.0 |

## Sensitivity and specificity of novel qPCR assay

Quantitative PCR was conducted in 20 μL reactions which included forward and reverse primers at a final concentration of 250 nM, 1x SsoAdvanced Universal SYBR Green Supermix (Bio-Rad, Australia) and 2 μL DNA template. PCR conditions were an initial denaturation of 98˚C for 3 min, followed by 40 cycles of denaturation at 98˚C for 10 s and annealing/extension at 56˚C for 20 s, and a final melt curve of 56–95˚C at 0.5˚C increments. A synthetic positive control was constructed using a pMG-Amp vector (Macrogen, South Korea) containing the target region and flanking sequence (166 bp) to generate a standard curve and calculate the limit of detection (LOD). The efficiency and LOD were calculated using a minimum of four replicates of $1 \times 10^8$ to $1 \times 10^4$ copies/μL, and ten replicates of $1 \times 10^4$, $1 \times 10^3$, 50, 25, 10, 5 and 1 copy/μL (S1 Table in S1 File).

Upon establishment of LOD and standard curve of the qPCR assay, we tested samples in duplicate using the reaction conditions as outlined above. Test qPCRs were conducted alongside a standard curve comprising the synthetic positive control at 10-fold dilutions ($1 \times 10^7$ to 1 copy/ μL) for quantification and 2 μL of PCR grade water as a no template control. Previously published primers amplifying β-actin [31] were used as a quality check for DNA quantity and quality and presence of inhibitors. A sample was considered positive if both duplicates at single or multiple anatomical swab sites successfully amplified β-actin and achieved a cycle threshold (Ct) of < 37 (equivalent to 12 copies, see Results) for PhaHV-1 and a melt at 81–81.5˚C. Any sample with discordant results between duplicates was retested and samples that failed to amplify β-actin were re-run at 1:10 dilution.

To confirm our qPCR specificity, we tested DNA extracts from KoRV, *C. pecorum*, Macropodid herpesvirus 1 (MaHV-1) and PhaHV-2 positive samples. In addition, a subset of samples (N = 12) that produced a positive PhaHV-1 result (according to criteria above), together with amplicons from the three known positive PhaHV-1 DNA samples and PCR product amplified from the PhaHV-1 synthetic positive control were sent to Macrogen, South Korea for by Sanger bidirectional sequencing. Resulting forward and reverse sequences were used to make a consensus sequence for each sample and then subjected to BLASTn using the NCBI database [30] to check amplicon sequence identity.

## PhaHV-1 and -2 point-of-care test (LAMP) development and evaluation

In order to design LAMP primers for PhaHV-1 and PhaHV-2 *dpol* gene, three different LAMP primer design software were used: licensed LAMP Designer 1.15 software (Premier Biosoft, CA, USA), open-source Primer Explorer v5 (Fujitsu Limited, Japan) and whole Genome based LAMP primer designer (GLAPD) [32]. As for the qPCR assay, no appropriate primers were predicted for PhaHV-2 so further work was conducted for PhaHV-1 *dpol* only. Primers were evaluated for sequence specificity *in silico* using PrimerBLAST [30] and primer characteristics using online tools OligoEvaluator (http://www.oligoevaluator.com) and OligoAnalyzer [29]. Upon preliminary testing of primers using a Genie III fluorometer (to evaluate time to amplify, amplification and melt curves for each set using DNA extracted from PhaHV-1 culture), we selected the most optimal primer set for PhaHV-1, designed by GLAPD (Table 2). The outer F3/B3 primers amplify a 216 bp fragment of the PhaHV-1 *dpol* gene, near the 5' end and distinct from the region targeted by our qPCR primers (S1 Fig in S1 File).

## PhaHV-1 isothermal assay evaluation

All LAMP assays were performed in 25 μL reaction volumes, consisting of 15 μL Isothermal Master Mix ISO001 (Optigene, UK), 5 μL primer mix (at 0.2 μM F3 and B3, 0.8 μM FIP and BIP, and 0.4 μM LF and LB as per primer mix for regular reactions), and 1 μL template (for

**Table 2. LAMP primers for PhaHV-1 developed in this study.**

| Primer | Sequence 5'-3' | Length |
|---|---|---|
| F3 | ACTTTGTTCCAGGACGTCTC | 20 bp |
| B3 | GTTTCCTTGGATGGGCTCC | 20 bp |
| FIP | GCTGCCCCAGATGTTCTGAAGCCCCAAACACCCTACCCAATC | 42 bp |
| BIP | TCTTCCCCAGTGACCATGGTCCAGCTCCTTCCCCCAGAATTA | 42 bp |
| LF | AGTTAGGCACCAATCGCGTGAT | 22 bp |
| LB | ATGGAAGACAAATTTACACCC | 21 bp |
| F2 | CCCAAACACCCTACCCAATC | 20 bp |
| F1c | GCTGCCCCAGATGTTCTGAAGC | 22 bp |
| B2 | AGCTCCTTCCCCCAGAATTA | 20 bp |
| B1c | TCTTCCCCAGTGACCATGGTCC | 22 bp |

sensitivity assays only) or 5 μL template (for specificity and sample testing assays); run at 65˚C for 30 min followed by denaturation step of 98–80˚C at a rate of 0.05˚C/s in the Genie III real-time fluorometer (Optigene, UK) to create a high-resolution melt curve. Positive (PhaHV-1 DNA) and negative (MilliQ water) controls were included in each assay.

To evaluate analytical sensitivity (LOD) of the PhaHV-1 isothermal assay, we used 1 μL of quantified PhaHV-1 DNA (as determined by qPCR of PhaHV-1 DNA extracted from a culture against the standard curve generated from serially diluted synthetic positive control described above) as a template in serial dilutions from 500 copies/reaction (tested in triplicate), and 100, 50, 10, 1 and 0 copies/reactions (all tested as five replicates). Each serial dilution set was prepared separately from the same quantified PhaHV-1 DNA, with dilutions tested in separate runs by the same operator (S2 Table in S1 File). The specificity of the newly developed PhaHV-1 LAMP assay was evaluated against a range of previously screened and confirmed DNA extracts from KoRV, *C. pecorum*, MaHV-1 and PhaHV-2 positive samples (S3 Table in S1 File).

## Isothermal testing of DNA and rapidly processed swab samples

The newly developed PhaHV-1 LAMP assay was evaluated against 70 DNA samples, previously extracted from ocular, nasal, urogenital and rectal mucosal swabs from 38 koalas, including asymptomatic individuals and with those signs of chlamydial and/or other clinical disease, available in the University of the Sunshine Coast sample collection. To evaluate the use of rapidly processed swabs in the PhaHV-1 assay, we used a total of 36 mucosal swabs (Minitip Rayon dry Aluminium shaft (Copan, Brescia, Italy) obtained from 16 ocular, 15 urogenital, one nasal and four rectal sites from 17 koalas. Swabs were processed in 300 μL of water by vortexing for 45 sec and heat lysis at 90˚C for 10 min, as previously described [26, 33]. All swabs were collected by qualified veterinarians as a part of routine health and diagnostic investigations. The testing and use of these swabs were approved by University of the Sunshine Coast Animal Ethics approval exemption (ANE2057).

All LAMP reactions were run as described above, with 5 μL of the DNA samples or swab suspensions used as template in a reaction. In these assays, a swab suspension or DNA sample was considered negative if amplification time was > 27.45 min with no or below threshold melt, as this was considered below detection level. Inhibition of amplification in swab suspensions was also assessed with 'spiking' experiments, wherein we added 1000 copies of PhaHV-1 DNA/reaction to 14 aliquots (50 μL) of selected swab suspensions that tested negative. The remaining volume of the swab suspension was used for DNA extraction using QiaAMP DNA mini kit, as per manufacturer instructions (Qiagen, Australia). Similarly, to assess whether the extracted swab

DNA may carry over inhibitors, we then spiked 50 μL of the DNA with 1000 copies of PhaHV-1 DNA per aliquot. Sample quality was evaluated using koala β-actin qPCR [31].

In order to evaluate LAMP relative to the newly developed PhaHV-1 qPCR assay, all 106 DNA samples consisting of 70 previously extracted and 36 freshly extracted DNA samples from mucosal swabs (separate to the sample set used in qPCR assay development) were also tested in duplicate with the PhaHV-1 qPCR assay as described above. Samples with a Ct < 37 or without a melt curve consistent with the positive controls, were considered negative.

### Statistical analyses

The performance of the tests conducted on the same samples was evaluated using EpiTools online epidemiological calculators using diagnostic test evaluation modules [34] with calculated Kappa and positive and negative agreement proportions, as well as McNemar's Chi-squared value with 95% confidence. Agreement analysis was considered more appropriate than relative sensitivity and specificity, as both tests were novel and, therefore, neither could be designated as the gold standard. It is suggested the Kappa value be interpreted as follows: values ≤0 as indicating no agreement and 0.01–0.20 as none to slight, 0.21–0.40 as fair, 0.41–0.60 as moderate, 0.61–0.80 as substantial, and 0.81–1.00 as almost perfect agreement.

## Results

### qPCR optimisation

PhaHV-1 qPCRs consistently resulted in melt temperatures of 81–81.5˚C (S2a Fig in S1 File) and efficiencies ranging from 90.1–106.7%. The limit of detection was determined as 12 copies per reaction (corresponding to a Ct-value of 37.5, S2b Fig and S1 Table in S1 File), with a quantifiable range over six 10-fold serial dilutions. Target specificity of the newly designed primers was confirmed by successful amplification of PhaHV-1 DNA and no amplification for other bacterial or viral DNA tested. BLAST results from sequenced PCR products all exclusively aligned to phascolarctid gammaherpesvirus 1 *dpol* with 98–100% sequence identity.

### Determining best sample for evaluating PhaHV-1 load/detection via qPCR

Of the 47 koalas tested using paired DNA extracts from swab samples, 27 (57%) were positive for PhaHV-1 as indicated by at least one anatomical site and 20 koalas were negative at all anatomical sites (S4 Table in S1 File). Initial investigation of optimal sample type for detection of PhaHV-1 infection indicated that of 47 paired sets of DNA extracted from oropharyngeal (Oro/Phx) and urogenital (UGT) swab samples, 12 koalas were positive for PhaHV-1 at both and 20 were negative at both. Ten koalas were positive at Oro/Phx and negative at UGT swabs and five koalas were negative at Oro/Phx and positive at UGT swabs. Of the six ocular swab results available from these animals, five were in agreement to their matched UGT and Oro/Phx sample results (1 positive and 4 negative) and one ocular swab was negative while the paired UGT and Oro/Phx samples were positive.

Of the 29 scat samples, seven were positive for PhaHV-1 in agreement with at least one swab sample from the same animal. There were no false positives results from scat samples, i.e., all scats were negative from 13 koalas whose swabs were also negative. There were nine scat samples that were negative where at least one swab sample from the same koala was positive. This corresponds to a specificity of 100% and a sensitivity of 44% when assessing this sample type.

## PhaHV-1 LAMP assay design, sensitivity and specificity

During preliminary assay design, we selected three primers sets, one of each designed by the three different LAMP primer tools. Although all three primer sets amplified PhaHV-1 DNA and no non-specific amplification was observed, primers designed with a GLAPD tool were selected for further assay optimisation and testing. This primer set was most efficient in achieving amplification of low DNA copy number template (added as 5 or 1 μL in reaction) in comparison to the other two primer sets (ie. the remaining sets had low amplification efficiency and were only able to amplify high DNA copy samples).

The limit of detection of the PhaHV-1 LAMP assay was conservatively estimated to be 10 genome copies/reaction (5/5 replicates, 100% detection) (S2 Table in S1 File). Similarly to the qPCR assay, target specificity of the newly designed primer set was confirmed by successful amplification of PhaHV-1 DNA, tested individually and/or as a mixture with other non-targeted DNA, and no amplification for other bacterial or viral DNA tested (S3 Table in S1 File).

## PhaHV-1 detection in DNA and rapidly processed clinical swabs using isothermal assays

Comparisons between the qPCR assay and the LAMP using 70 DNA samples, demonstrated 94.29% overall agreement between the two assays, with Kappa values of 0.8113 indicating an almost perfect agreement (Table 3, S5 Table in S1 File). Discrepant results were noted for four samples, where sample K4 OCC was positive by LAMP but negative by qPCR, and remaining three samples (K5 REC, K8 UGT, and K30 OCC) were negative by LAMP but positive by qPCR assay (S5 Table in S1 File).

For the 36 rapidly processed swab suspensions, the overall agreement between the PhaHV-1 LAMP and the qPCR assay was 94.44% with Kappa values of 0.7118 indicating substantial agreement (Table 4). Discrepant results were noted for two samples, where sample K6 REC was highly positive by LAMP but negative by qPCR, and sample K16 E was negative by LAMP but weakly positive by qPCR with an average Ct of 34.56 (S6 Table in S1 File).

## Testing for inhibition of amplification in rapidly processed samples

No PhaHV-1 was detected by LAMP in three of 14 "spiked" rapidly processed swab suspensions (samples K1 REC, K2 REC, and K16 UGT; S6 Table in S1 File), indicating the potential presence of inhibitors in these samples. When testing the DNA extracted from the same "spiked" swabs using qPCR, only one sample (K2 REC) was inhibited, with no PhaHV-1 DNA

**Table 3. Comparison of PhaHV-1 LAMP assay to the PhaHV-1 reference qPCR assay when testing DNA samples.**

| Test | Reference PhaHV-1 qPCR | | |
|---|---|---|---|
| **PhaHV-1 LAMP** | Positive | Negative | Total |
| Positive | 11 | 1 | 12 |
| Negative | 3 | 55 | 58 |
| Total | 14 | 56 | 70 |
| Overall agreement[a] | 94.29% (95% CI: 0.8601–0.9842) | | |
| | 84.62% PA (95% CI: 0.6972–0.9951) | | |
| | 96.49% NA (95% CI: 0.9305–0.9993) | | |
| McNemar's Chi sq (pChi sq) | 0.25 (0.6171) | | |
| Kappa (0.95 CI, pKappa) | 0.8113 (0.6336–0.9891; 0) | | |

[a] positive agreement (PA) and negative agreement (NA) are also outlined.

**Table 4.  Comparison of the PhaHV-1 LAMP testing of rapidly processed swabs to the PhaHV-1 reference qPCR testing of DNA extracted from the same swab suspensions.**

| Test | Reference PhaHV-1 qPCR | | |
|---|---|---|---|
| **PhaHV-1 LAMP** | Positive | Negative | Total |
| Positive | 3 | 1 | 4 |
| Negative | 1 | 31 | 32 |
| Total | 4 | 32 | 36 |
| Overall agreement[a] | 94.44 (95% CI: 0.8134–0.9932) | | |
|  | 75% PA (95% CI: 0.4145–1.0855) | | |
|  | 96.88% NA (95% CI: 0.9255–1.0120) | | |
| McNemar's Chi sq (pChi sq) | 0.5 (0.4795) | | |
| Kappa (0.95 CI, pKappa) | 0.7188 (95% CI: 0.3497–1.0878; 0) | | |

detected; PhaHV-1 DNA was detected in the DNA extracted from the remaining two "spiked" swab samples inhibited in LAMP (K1 REC and K16 UGT), but with late amplification (S6 Table in S1 File).

## Discussion

We have developed and optimised two new assays for detection of PhaHV-1 in koalas. The qPCR assay provides a high-throughput, specific, and sensitive diagnostic test that can quantify viral load; to be used in research settings alongside existing assays for chlamydia and KoRV diagnostics. This will allow further research on the distribution and prevalence of PhaHV-1 infection as well as increasing our understanding of the dynamics of co-infections. The LAMP assay provides a robust, rapid, point-of-care test, with applications in clinical and field settings. This will aid rapid diagnostics to guide quarantine of koalas in care. The use of both assays will enable vital research into the distribution, prevalence and significance of PhaHV-1 infection in the endangered koala populations of NSW, Qld and the ACT, where there is currently no information on this pathogen.

We recommend multiple samples from different anatomical sites from the same animal be used to maximise sensitivity of PhaHV-1 detection. In our study, PhaHV-1 detection by qPCR was slightly more sensitive in oropharyngeal swabs compared to urogenital swabs. Studies are equivocal as to whether detection is more likely in oropharyngeal or urogenital swabs [10, 11], and larger sample sizes may be needed to resolve this question. DNA extracted from scat was not a reliable sample for PhaHV-1 detection using our assay. We cannot conclusively determine whether the assays we have developed are detecting all infected animals or only lytic infections, as the virus in latent form may be present in circulating B cells [14]. Regardless, the ability to rapidly quantify viral load will advance research into the pathogenesis of PhaHV-1 infection, its relationship to stress and disease and its significance amongst other co-infections.

In this study, we also demonstrated the successful use of rapid isothermal assay of rapidly processed swabs to detect PhaHV-1 infections, supporting the proof of concept that rapid isothermal diagnostics can be applied at the koala and/or other veterinary clinical setting [33, 35, 36]. Similar to *Chlamydia* isothermal assays [26, 27], the PhaHV-1 isothermal assay performed in the Genie III fluorometer provides straightforward interpretation of results, with positive results being distinctly characterised by amplification time and specific melt curves. Although LAMP is considered robust to PCR inhibition, the presence of PCR inhibitors, from the sample itself or introduced during sample processing, can produce false-negative results [37] and we found this to be the case. We observed better congruence between LAMP and the reference qPCR when testing kit-extracted DNA samples; reduced sensitivity and increased inhibition of

LAMP was observed when testing rapidly processed swab samples. For example, two rectal (K1 REC and K2 REC) and a UGT (K16 UGT) swabs perhaps contained inhibitors and did not amplify in LAMP and qPCR (K2 REC) or produced late amplification (K1 RE, K16 UGT) on qPCR after DNA extraction. Nevertheless, rapid swab processing and crude DNA extraction can be improved using commercially available DNA lysis buffers and, as recommended above, swabs from multiple anatomical sites from the same animal should be tested. In this study, we only tested previously collected mucosal swabs, a further investigation testing fresh swabs, comparing sample storage methods and improved sample lysis is required to fully harness the potential of this technology.

The assays we have developed will support the National Koala Monitoring Program and state-based monitoring programs, translocation risk assessment, welfare and biosecurity in rehabilitation, as well as implementation of the National Koala Disease Risk Assessment [3]. Future work can develop rapid diagnostics for PhaHV-2, but this will first require more sequence information. The PhaHV-1 assays we have developed are now being made available in research and clinical settings, enabling collaborative research across institutions to better define the prevalence and impacts of PhaHV-1 infections.

## Supporting information

**S1 File. Contains all supporting tables and figures.**
(DOCX)

## Acknowledgments

The authors would like to thank Australia Zoo Wildlife Hospital, Friends of the Koala and Port Macquarie Koala Hospital for their contributions to sampling and Shannon Taylor for assisting with DNA extractions.

## Author Contributions

**Conceptualization:** Martina Jelocnik, Paola K. Vaz, Joanne M. Devlin, Damien P. Higgins.

**Data curation:** Belinda R. Wright, Martina Jelocnik, Andrea Casteriano, Yasmine S. S. Muir, Paola K. Vaz, Damien P. Higgins.

**Formal analysis:** Belinda R. Wright, Martina Jelocnik, Damien P. Higgins.

**Funding acquisition:** Damien P. Higgins.

**Investigation:** Belinda R. Wright, Martina Jelocnik, Andrea Casteriano, Alistair R. Legione, Paola K. Vaz, Joanne M. Devlin, Damien P. Higgins.

**Methodology:** Belinda R. Wright, Martina Jelocnik, Andrea Casteriano, Yasmine S. S. Muir, Alistair R. Legione, Paola K. Vaz, Joanne M. Devlin, Damien P. Higgins.

**Project administration:** Damien P. Higgins.

**Resources:** Martina Jelocnik, Paola K. Vaz, Damien P. Higgins.

**Supervision:** Andrea Casteriano, Joanne M. Devlin, Damien P. Higgins.

**Validation:** Belinda R. Wright, Martina Jelocnik.

**Writing – original draft:** Belinda R. Wright, Martina Jelocnik.

**Writing – review & editing:** Belinda R. Wright, Martina Jelocnik, Andrea Casteriano, Yasmine S. S. Muir, Alistair R. Legione, Paola K. Vaz, Joanne M. Devlin, Damien P. Higgins.

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
