## [Decision Letter · Decision Letter 0]

29 Mar 2023

PONE-D-23-04405Development of diagnostic and point of care assays for a gammaherpesvirus infecting koalas.PLOS ONE

Dear Dr. Wright,

Thank you for submitting your manuscript to PLOS ONE. After careful consideration, we feel that it has merit but does not fully meet PLOS ONE’s publication criteria as it currently stands. Therefore, we invite you to submit a revised version of the manuscript that addresses the points raised during the review process.

We look forward to receiving your revised manuscript.

Kind regards,

Jiro Yasuda, PhD, DVM

Academic Editor

PLOS ONE

Journal Requirements:

   "This study was funded by the Australian Department of Agriculture, Water and the Environment Bushfire Recovery Multiregional Species Program. BRW is supported by the NSW Wildlife Information Rescue and Education Service (WIRES)." 

Reviewers' comments:

Reviewer's Responses to Questions

**Comments to the Author**

1. Is the manuscript technically sound, and do the data support the conclusions?

Reviewer #1: Yes

Reviewer #2: Partly

2. Has the statistical analysis been performed appropriately and rigorously? 

Reviewer #1: Yes

Reviewer #2: No

3. Have the authors made all data underlying the findings in their manuscript fully available?

Reviewer #1: Yes

Reviewer #2: Yes

4. Is the manuscript presented in an intelligible fashion and written in standard English?

Reviewer #1: Yes

Reviewer #2: Yes

5. Review Comments to the Author

Reviewer #1: The paper "Development of diagnostic and point of care assays for a gammaherpesvirus infecting

koalas" is a straightforward diagnostic assay development that appears to have run all appropropriate controls robustly and should provide useful tools for ongoing diagnostic and epidemiology work in this species, the assays are badly needed for field studies and should be useful going forward and appear to be robust in terms of sensitivity and specificity. I have only minor criticisms (mostly to do with data display rather than any real criticism of the methodology. The use of taqman qPCR probes is generally preferred to sybr green assays for diagnostics (to increase specificity), however your melt curve and sequencing data on field samples shows your assay is robust enough to run without the probes so I am happy with this set up and assay.

I would in general prefer to see the standard curve (presented in a standard log concentration vs CT value) as well as your melt curves presented for a new assay to allow assessment of the robustness of the assay (and also for anyone else using your assay to be able to compare their systems performance to yours). I am not very clear what your cut off graph is trying to show (are the values below 10 the ones with dodgy melt curves or where duplicates did not amplify? ) unfortunately the version I can see doesn't have the figure legends on it.

There has been some debate recently over the use of beta actin as a standard for koala samples as it is not a single copy number gene. This is however a minor issue for quality control, just something to keep in mind if doing virus load/estimated cell count type quantifications (which are probably not relevant anyway for swab samples as you can't tell if this is cell associated or free virus)

Reviewer #2: The authors reported the development of qPCR and LAMP assays for the detection of Phascolarctid gammaherpesvirus-1 (PhaHV-1) infecting koalas in southern Australia. PhaHV-1 has been detected in koalas with Chlamydia pecorum or koala retrovirus. The virus may also cause latent infection in lymphocytes, and then be associated with lymphoproliferative disorders in immunocompromised individuals. The manuscript was well written and the interpretation of the data was fairly discussed. I hope that the following comments will help to improve the manuscript and get it published in the journal.

Table 3

I suppose that the indicated values of PA and NA are not correct. According to the numbers described in this 2x2 table, these values should be as following;

PA(sensitivity)= 11/14= 78.57%

NA(specificity)=55/56=98.21%

Please confirm your calculated results.

Discussion

In the second paragraph, the authors suggested that virus detection was slightly more sensitive in oropharyngeal swabs than in urogenital swabs. For this discussion, I recommend that the authors prepare a figure plotting the Ct values of the respective samples from oropharyngeal and urogenital swabs and demonstrate a significant difference between the Ct values of the two groups.

Table S3

Please describe the name of the strain (if available) and amount of bacterial or viral DNA used here.

Table S6

Please explain the abbreviation of NA in the footnote.

6. PLOS authors have the option to publish the peer review history of their article (what does this mean?). If published, this will include your full peer review and any attached files.

Reviewer #1: No

Reviewer #2: No

---

## [Author Response · Author response to Decision Letter 0]

20 Apr 2023

Reviewers' comments:

Reviewer's Responses to Questions

Comments to the Author

1. Is the manuscript technically sound, and do the data support the conclusions?

Reviewer #1: Yes

Reviewer #2: Partly

2. Has the statistical analysis been performed appropriately and rigorously?

Reviewer #1: Yes

Reviewer #2: No

3. Have the authors made all data underlying the findings in their manuscript fully available?

Reviewer #1: Yes

Reviewer #2: Yes

4. Is the manuscript presented in an intelligible fashion and written in standard English?

Reviewer #1: Yes

Reviewer #2: Yes

5. Review Comments to the Author

Reviewer #1: The paper "Development of diagnostic and point of care assays for a gammaherpesvirus infecting koalas" is a straightforward diagnostic assay development that appears to have run all appropropriate controls robustly and should provide useful tools for ongoing diagnostic and epidemiology work in this species, the assays are badly needed for field studies and should be useful going forward and appear to be robust in terms of sensitivity and specificity. I have only minor criticisms (mostly to do with data display rather than any real criticism of the methodology. The use of taqman qPCR probes is generally preferred to sybr green assays for diagnostics (to increase specificity), however your melt curve and sequencing data on field samples shows your assay is robust enough to run without the probes so I am happy with this set up and assay.

I would in general prefer to see the standard curve (presented in a standard log concentration vs CT value) as well as your melt curves presented for a new assay to allow assessment of the robustness of the assay (and also for anyone else using your assay to be able to compare their systems performance to yours). I am not very clear what your cut off graph is trying to show (are the values below 10 the ones with dodgy melt curves or where duplicates did not amplify? ) unfortunately the version I can see doesn't have the figure legends on it.

Response- We have amended figure S2 (now Fig S2b) to present the standard concentration on a log scale and omitted the cut off red line as it appeared to cause confusion. This line depicted the limit of detection but as this is already outlined in the manuscript text (line 228) it is unnecessary. We have also added the melt curves to this figure as Fig S2a to allow readers to use as a reference when using the assay.

There has been some debate recently over the use of beta actin as a standard for koala samples as it is not a single copy number gene. This is however a minor issue for quality control, just something to keep in mind if doing virus load/estimated cell count type quantifications (which are probably not relevant anyway for swab samples as you can't tell if this is cell associated or free virus)

Response- We agree with the reviewer that beta actin may have some issues as a reference gene for koalas but as this gene has widespread use across many assays and laboratories, it is the most appropriate reference gene available. Furthermore, koala beta actin is also commonly used as a control to check that mucosal swabs were collected properly (i.e cells present), which is relevant for herpesvirus, Chlamydia and KORV testing. 

Reviewer #2: The authors reported the development of qPCR and LAMP assays for the detection of Phascolarctid gammaherpesvirus-1 (PhaHV-1) infecting koalas in southern Australia. PhaHV-1 has been detected in koalas with Chlamydia pecorum or koala retrovirus. The virus may also cause latent infection in lymphocytes, and then be associated with lymphoproliferative disorders in immunocompromised individuals. The manuscript was well written and the interpretation of the data was fairly discussed. I hope that the following comments will help to improve the manuscript and get it published in the journal.

Table 3

I suppose that the indicated values of PA and NA are not correct. According to the numbers described in this 2x2 table, these values should be as following;

PA(sensitivity)= 11/14= 78.57%

NA(specificity)=55/56=98.21%

Please confirm your calculated results.

Response- We have rechecked the results using statistical calculator for test comparisons, Epitools online using https://epitools.ausvet.com.au/comparetwotests module. As advised by a biostatistician, this module is to be used when neither test is “Gold standard reference” test (like our newly developed qPCR and LAMP) and that we should evaluate agreement.

You are right about the values you have recalculated for sensitivity/specificity; however, we were advised that these are used when disease status is explicitly known and when one of the tests is fully validated “Gold standard reference”. When using such parameters, we achieve the same results using Epitools online (https://epitools.ausvet.com.au/testevaluation) as well as MedCalcs (https://www.medcalc.org/calc/diagnostic_test.php). Please see table below. 

In our study with newly developed assays and observation that PhaHV-1 can be found in asymptomatic animals, we feel its more appropriate the use agreement indices as above. 

 Point Estimate Lower 95% CL Upper 95% CL

Sensitivity 0.7857 0.492 0.9534

Specificity 0.9831 0.9091 0.9996

Likelihood ratio +ve 46 

Likelihood ratio -ve 0.218 

Discussion

In the second paragraph, the authors suggested that virus detection was slightly more sensitive in oropharyngeal swabs than in urogenital swabs. For this discussion, I recommend that the authors prepare a figure plotting the Ct values of the respective samples from oropharyngeal and urogenital swabs and demonstrate a significant difference between the Ct values of the two groups.

Response- We thank the reviewer for this suggestion however we were not evaluating differences in viral load in the different swab types but simply the ability to detect PhaHV-1 in each swab type. At present, such a plot would only be able to compare ct values of the 12 samples that were positive at both swab type, and it is certainly one of the next comparisons that could be done using larger number of various samples. We have not stated that the difference is significant and are here in the discussion simply stating that for accurate detection of PhaHV-1, multiple sample types should be used (line 309).

We have reworded the sentence at Line 301 for clarity: “ In our study, PhaHV-1 DNA detection by qPCR in oropharyngeal swabs was somewhat more achievable compared to urogenital swabs”. 

Table S3

Please describe the name of the strain (if available) and amount of bacterial or viral DNA used here.

Response- We have now added these details to table S3. The strains names are: PhaHV1 is 36M/11, MaHV1 it’s V3076/08 and for C. pecorum is Marsbar_2018 koala isolate. No strain name for KORV, it’s a DNA samples confirmed positive for KORV and C. pecorum and negative for PhaHV-1, available in our catalogue. A 5µl template was added in the reaction for all testing, as recommended by the manufacturer.

Table S6

Please explain the abbreviation of NA in the footnote.

Response-NA refers to no amplification achieved. We have added this in footnote.

---

## [Editor Report · Decision Letter 1]

15 May 2023

Development of diagnostic and point of care assays for a gammaherpesvirus infecting koalas.

PONE-D-23-04405R1

Dear Dr. Wright,

We’re pleased to inform you that your manuscript has been judged scientifically suitable for publication and will be formally accepted for publication once it meets all outstanding technical requirements.

Kind regards,

Jiro Yasuda, PhD, DVM

Academic Editor

PLOS ONE

---

## [Editor Report · Acceptance letter]

23 May 2023

PONE-D-23-04405R1 

Development of diagnostic and point of care assays for a gammaherpesvirus infecting koalas. 

Dear Dr. Wright:

I'm pleased to inform you that your manuscript has been deemed suitable for publication in PLOS ONE. Congratulations! Your manuscript is now with our production department. 

Kind regards, 

on behalf of

Professor Jiro Yasuda 

Academic Editor

PLOS ONE